

# Performance evaluation of throughput-aware framework for ensemble data assimilation: The case of NICAM-LETKF

H. Yashiro[1], K. Terasaki[1], T. Miyoshi,[1,2,3] and H. Tomita[1]

[1]RIKEN Advanced Institute for Computational Science, Kobe, Japan
[2]Application Laboratory, Japan Agency for Marine-Earth Science and Technology, Yokohama, Japan
[3]University of Maryland

*Correspondence to*: Hisashi Yashiro, RIKEN/Advanced Institute for Computational Science, 7-1-26 Minatojima-minami-machi, Chuo-ku, Kobe, Hyogo 650-0047, Japan

Tel: +81-78-940-5731. E-mail: (h.yashiro@riken.jp)

**Abstract.** In this paper, we propose the design and implementation of an ensemble data assimilation (DA) framework for weather prediction at a high resolution and with a large ensemble size. We consider the deployment of this framework on the data throughput of file input/output (I/O) and multi-node communication. As an instance of the application of the proposed framework, a Local Ensemble Transform Kalman Filter (LETKF) was used with a Non-hydrostatic Icosahedral Atmospheric Model (NICAM) for the DA system. Benchmark tests were performed using the K computer, a massive parallel
supercomputer with distributed file systems. The results showed an improvement in total time required for the workflow as well as satisfactory scalability of up to 10 K nodes (80 K cores). With regard to high-performance computing systems, where data throughput performance increases at a slower rate than computational performance, our new framework for ensemble DA systems promises drastic reduction of total execution time.

## 1 Introduction

Rapid advancements in high-performance computing (HPC) resources in recent years have enabled the development of atmospheric models to simulate and predict the weather at high spatial resolution. For effective use of massive parallel supercomputers, parallel efficiency becomes a common but critical issue in weather and climate modeling. Scalability for several large-scale simulations has been accomplished to a certain extent thus far. For example, the Community Earth System Model (CESM) performs high-resolution coupled climate simulations by using over 60 K cores of an IBM Blue
Gene/P system (Dennis et al., 2012). Miyamoto et al. (2013) generated the first global sub-km atmosphere simulation by using a Non-hydrostatic Icosahedral Atmospheric Model (NICAM) with 160 K cores of the K computer.

Climate simulations at such high resolutions need to be able to handle the massive amounts of input/output (henceforth, I/O) data. Since the throughput of file I/O is much lower than that of the main memory, I/O performance is important to maintaining the scalability of the simulations as well as guaranteeing satisfactory computational performance. Parallel I/O is



necessary to improve the total throughput of I/O. In order to improve performance, a few libraries have been developed for climate models, e.g., the application-level parallel I/O (PIO) library, which was developed (Dennis et al., 2011) and applied to each component model of the CESM. The XML I/O server (XIOS, http://forge.ipsl.jussieu.fr/ioserver) was used in European models, such as the EC-EARTH (Hazeleger et al. 2010). XIOS distinguishes the I/O node group from the

simulation node group, and asynchronously transfers data for output generated by the latter group to the former. With the development of models at increasing spatial resolution, the use of parallel I/O libraries will become more common.

In addition to the simulation that is generated, the performance of the data assimilation (DA) system plays an important role in the speed of numerical weather prediction. Many DA systems have been developed, e.g., variational methods, Kalman filters, particle filters, etc. In particular, two advanced DA methods—the four-dimensional variational (4D-Var) method

(Lorenc, 1986), and the ensemble Kalman filter (EnKF, Evensen, 1994; 2003)—are used at operational forecasting centers. Hybrid ensemble/4D-var systems have also been recently developed (Clayton et al., 2013). 4D-Var systems require an adjoint model that relies heavily on the simulation model. On the contrary, DA systems using the EnKF method are independent of the model. Ensemble size is a critical factor in obtaining statistical information regarding the simulated state in an ensemble DA system. Miyoshi et al. (2014; 2015) performed 10,240-member EnKF experiments, and proposed that the

typical choice of an ensemble size of approximately 100 members is insufficient to capture the precise probability density function and long-range error correlations. Thus, it is reasonable to increase not only the resolution of the model, but also its ensemble size in accordance with performance enhancement yielded by supercomputers. However, this enhancement in model resolution and ensemble size leads to a tremendous increase in total data input and output. For example, prevalent DA systems operating at high resolution with a large numbers of ensemble members require terabyte-scale data transfer between

components. In the future, the volume of data in large-scale ensemble DA systems is expected to reach the petabyte scale.

In such cases, data movement between the simulation model and the ensemble DA systems will become the most significant issue. This is because data distribution patterns for inter-node parallelization in the two systems are different. The processes of a simulation model share all global grids of a given ensemble member. On the contrary, the DA system requires all ensemble members for each process. Even if the simulation model and the DA system use the same processes, the data

layout in each is different and, hence, needs to be altered between them. Thus, a large amount of data exchange through inter-node communication or file I/O is required. This problem needs to be addressed in order to enhance the scalability of the ensemble DA system.

As described above, data throughput between model simulations and ensemble DA systems becomes much larger than that for single atmospheric simulations. We are now confronted with the problem of data movement between the two components.

This study aims to investigate the performance of ensemble DA systems by focusing on reducing data movement. NICAM (Satoh et al., 2014) and Local Ensemble Transform Kalman Filter (LETKF) (Hunt et al., 2007) were used as reference cases for the model and the DA system, respectively. In Section 2, we summarize the design and implementation of the conventional framework for ensemble DA systems, and illuminate the problem from the perspective of data throughput. To solve the problem, we propose our framework for DA systems in Section 3. In order to test the effectiveness of our



framework, we describe performance and scalability in the case of NICAM and LETKF on the K computer, which has a typical mesh torus topology for inter-node communication, in Section 4. We summarize and discuss the results in Section 5.

## 2 NICAM–LETKF DA system

NICAM (Satoh et al., 2014) is a global non-hydrostatic atmospheric model developed mainly at the Japan Agency for Marine-Earth Science and Technology, the University of Tokyo, and RIKEN Advanced Institute for Computational Science. With the aid of state-of-the-art supercomputers, NICAM has been contributing to atmospheric modeling at high resolutions. The first global simulations with a 3.5-km horizontal mesh were carried out on the Earth Simulator. The simulations showed a realistic multi-scale cloud structure (Tomita et al. 2005; Miura et al. 2007b). The K computer allowed many more

simulations at the same or higher resolutions. Miyakawa et al. (2014) showed using several case studies that the skill score of the Madden–Julian Oscillation (MJO) (Madden and Julian 1971, 1972) improved by using a convection-resolving model in comparison with other models. As a climate simulation, the 30-year AMIP-type simulation was conducted with a 14-km horizontal mesh (Kodama et al., 2015). The global sub-km simulation revealed that the essential change in convection statistics occurred at a grid spacing of approximately 2 km (Miyamoto et al. 2013). NICAM employs fully compressible non-

hydrostatic dynamics, where the finite volume method is used for discretization on the icosahedral grid system. The grid point method has the advantage of reducing data transfer between computational nodes over a spectral transform method, which requires global communication between nodes and constitutes one of the bottlenecks in a massively parallel machine. The LETKF (Hunt et al., 2007) is an advanced data assimilation method based on the local ensemble Kalman filter (LEKF; Ott et al. 2004), where the ensemble update method of the ensemble transform Kalman filter (ETKF; Bishop et al. 2001) is

applied to reduce computational cost. The LETKF has been coupled with a number of weather and climate models. For example, Miyoshi and Yamane (2007) applied the LETKF to the global spectral model AFES (Ohfuchi et al. 2004), Miyoshi et al. (2010) applied it to an operational global model developed by the Japan Meteorological Agency (JMA), and Miyoshi and Kunii (2012) constructed the WRF (Skamarock et al. 2005)–LETKF system. Kondo et al. (2009) were first to conduct simulation experiments under the perfect model scenario by using the LETKF with NICAM. Terasaki et al. (2015)

developed a NICAM–LETKF system for experiments with real data. In addition to its impressive physical performance, a reason for why many prevalent DA systems employ the LETKF lies in its massive parallel computation ability, where the analysis of calculation is separately executed for each grid. The NICAM–LETKF system is based on the code for the LETKF by Miyoshi (2005). Miyoshi and Yamane (2007) applied a parallel algorithm to the LETKF for efficient parallel computation, and Miyoshi et al. (2010) addressed load imbalance in the algorithm.

The following is devoted to an explanation of the current NICAM-LETKF and a clarification of the problem. Figure 1 shows a flow diagram of the DA system with the LETKF and an atmospheric model. In this DA system, three application programs are used: an atmospheric simulation model, a simulation-to-observation converter (henceforth, StoO), and the LETKF. These



programs are executed sequentially in a DA cycle. Most atmospheric models often use aggregated data for file I/O. This framework also assumes that each member has only a single file containing the simulated state. The numbers of computational nodes to be used are separately set for each program component. Since no component contains knowledge of the process used for file I/O, the output should be located in the shared file system; otherwise, the components cannot share

information with one another. The StoO program reads the simulation results $[X_f]$ and observation data $[y]$ as a first guess. The simulation results are diagnostically converted into observed variables $[H(x_f)]$. By using information regarding the horizontal and vertical locations, the model grid data are interpolated to data at the position of observation. Variable conversions, such as radiation calculations, are also applied when necessary. Following the conversion, the difference between the converted simulation results and the observations $[H(x_f)-y]$ is calculated for the output. The StoO program is

independently executed for each ensemble member. In the first version of the NICAM–LETKF system, raw simulation data on the icosahedral grid are once converted to fit the latitude–longitude grid. Following this interpolation, the StoO program generates variables at the observational point using another interpolation. Although this enables the use of existing DA code, the redundant interpolation incurs time and yields additional interpolation error. Terasaki et al. (2015) improved this by directly using data on the icosahedral grid for interpolation at the observation point, instead of using pre-converted data from

the icosahedral to the latitude-longitude grid system. The LETKF program reads the simulation and the results of the StoO. Processes equal in number to the ensemble size are selected to read the simulation results in parallel. Each selected process reads a member of the simulation result $[X_f]$ and distributes grid data to all other processes by scatter communication. Following the data exchange, the main computational part of the LETKF is separately executed in each process. The results are exchanged once again by gathering communication among all processes to generate the new initial grid states $[X_a]$ from

the selected processes in parallel.

The workflow described above has the following three bottlenecks:

1)      Limitation in the total throughput of I/O

2)      Collision of I/O requests due to a shared file system (FS)

3)      Global communication of large amounts of data

Improvement in the parallel efficiency of the LETKF has thus far been made from the viewpoint of computation (Miyoshi and Yamane 2007, Miyoshi et al., 2010). The three bottlenecks above are related to data movement. We discuss them in detail. First, the number of nodes for the input simulated state is limited to the number of ensemble members. With a simulation model of increasing resolution, the amount of output data increases. Nevertheless, the number of available nodes is limited to the ensemble size in the DA system. This limitation is due to the assumption that the model output is a single

file. As a result, the time to read grid data increases in the absence of scalability. Second, the use of a shared file system (FS) causes I/O communication to slow down. I/O performance is related not only to throughput, but also to inundation of the I/O request. Many HPC systems adopt distributed parallel FS, such as Lustre (http://lustre.org/), that enable parallel disk access and improve I/O throughput. However, a latent bottleneck in the metadata server occurs when a large number of processes simultaneously access the file system to write data. Third, global communication takes a long time with a large amount of



data. This problem becomes more serious in high-resolution and large ensembles. A greater number of grid data items take longer to distribute to all processes. The increase in the total number of processes also requires time to complete data exchange. Note that this is more or less true of any network topology. The scalability of the ensemble DA system on massive parallel supercomputers worsens due to these bottlenecks.

## 3 Proposed NICAM–LETKF framework

To solve the three problems with the current workflow stated and explained in Section 2, we design and implement a framework for the NICAM–LETKF system. The key concepts of data handling in the new framework are shown in Fig. 2. This framework is based on the I/O pattern of NICAM, which handles horizontally divided data such that each process separately reads and writes files. In an ensemble simulation, the total number of processes is equal to the horizontally

divided processes multiplied by the ensemble size. This is equal to the number of output files. Output data from each process is written to a local disk. We assume that this local disk is not shared by any other process. In this framework, we use the same number of processes in each of the three program components. All processes are used for I/O in every program. We use MPI_Alltoall to exchange grid data (we call this "shuffle") in StoO and the LETKF. The processes of ensemble members in the same positions in the grids are grouped for MPI communication. All ensemble members in the same local region are

included in the same group. This grouping can minimize the number of communication partners and reduce the total data transfer distance. We can hence avoid a global shuffle, which is the third problem with conventional frameworks. Following the computation of the LETKF, the data for analysis are shuffled again. Data for the next simulation are then transferred to the local disk, in the reverse order of the input stage.

The above concepts of the proposed framework can be applied to any simulation model. The model can use any grid system,

structured or unstructured. Based on these concepts, the method of implementation in NICAM–LETKF is a typical example of models with a structured and complicated grid system. NICAM adopts the icosahedral grid configuration, where the grids quasi-homogeneously cover the sphere, and are horizontally divided into groups called "regions" (Tomita et al., 2008). One or more regions are assigned to each process. The global grid is constructed by a recursive method (Tomita et al 2002, Stuhne and Peltier, 1999). The regions are also constructed with a rule similar to the recursive division method. Thus, the

structure of the local grids is kept in the region. We also adopt the same method for grid distribution in each shuffling group. Figure 3 shows the schematic picture of grid division. By using a mini-region as a unit, we can retain the grid mesh structure. This method is advantageous when we interpolate the grid data from the icosahedral grid system to the location of observation in StoO. However, this rule limits the available number of processes in the shuffling group, which is equal to the number of ensemble members. In the case of NICAM, there are $10 \times 4^n$ regions, where n is an integer greater than zero. We

can use a divisor of the total number of regions as the number of regions to assign to each process. The number of mini-regions depends on the number of regions in a process. For example, we can configure the horizontal grid as follows; the total number of regions is set to 160. Two regions are assigned to each process, and $16 \times 16$ grids are contained in each





region. At this setting, we can use 1, 2, 4, 8, 16, 32, 64, 128, 256, or 512 as ensemble size. We can choose any division method of a local grid group, but assign priority to the efficiency of interpolation calculation and load balancing in this study. In the proposed framework, only the master process of all MPI processes manages the I/O of the observation data and the results of the StoO. Global communication is used to broadcast and aggregate these data items. In this study, the size of these

data items is smaller than 50 MB, because of which the time needed for I/O and the communication of these data items is short. We leave issues arising from a large amount of observation data as part of future research, and reflect on it in our discussion.

## 4 Performance evaluation

In this section, we describe experiments to test the proposed framework on the K computer. This computer system is

equipped with both global and local FS. The user enters initial data from the global FS to the local FS through the staging process. The local FS in a node has a shared directory with all other nodes, and a local (rank) directory used only by the node. Although the shared directory allows all nodes to access one another, its throughput is degraded by the frequency of requests for I/O from them. On the contrary, the local directory can maximize the efficiency of total I/O bandwidth because any conflict in I/O between nodes is avoided by reducing the load on the metadata server. In our comparative case study, the old

framework used a shared directory in the conventional manner while the new framework used only the rank directory.

Table 1 summarizes the experimental setup: the resolutions, the number of ensembles, the number of processes, and so forth. As the observational data was assimilated into the results of the model, the NCEP PREPBUFR (available at http://rda.ucar.edu/data sets/ds337.0) observation dataset was used. Data thinning was applied according to a 112-km mesh, and the same number of total observations were used for all experiments. Covariance localization was adopted by using a

Gaussian function within 400 km in the horizontal and 0.2 $ln(p)$ in the vertical directions, where $p$ represents pressure. Note that the simulation with a 28-km mesh employed a more sophisticated cloud microphysics scheme.

Figure 4 shows the breakdown of elapsed time for a DA cycle in the case involving 256 members. The blue bar shows NICAM, whereas the green and red bars show StoO and the LETKF, respectively. The shaded part represents the time taken for communication and I/O. As reference, we confirmed that the 112 km mesh experiment took comparable times in model

simulation and data assimilation when using the 2,560 processes. The computation times for the StoO and the LETKF increased fourfold in the 56-km mesh experiments, as shown in Fig. 4. This was reasonable in light of the increase in horizontal resolution. On the contrary, the time required for the simulation increased almost eightfold. If we halve the grid spacing, we have to halve the Δt. Therefore the number of simulation steps doubles. A higher resolution incurred longer execution time than that needed for data assimilation. Thus, we need to increase the number of computation nodes to shorten

elapsed time. For example, as shown in Fig. 4, the number of nodes increased fourfold. Although we expected a fourfold reduction in time, the old framework could not attain effective reduction in data assimilation due to the bottleneck associated



with I/O and the communication components. By contrast, the proposed framework yielded the scalability in terms of computation, IO, and communication.

From the viewpoint of computational efficiency, we obtained 36 TFLOPS of sustained performance in a DA cycle with 10,240 processes that corresponded to 81,920 cores. The ratio of computation time to total time improved from 0.44 in the conventional framework to 0.76 in the proposed framework. From Fig. 4, we see that the computation time of the LETKF in the proposed framework increased in comparison with that in the old framework. This is because of load imbalance in the former. The number of observational data items used for data assimilation at each grid point depended on the localized radius in the LETKF and the spatial homogeneity of the observational data. The conventional framework can avoid this imbalance by shuffling the grid among all nodes (Miyoshi et al., 2010). However, the proposed framework cannot avoid it because each process manages its spatially localized region. In other words, the proposed framework reduces data movement at the cost of load balancing. We argue that this deficiency is not a problem in light of anticipated assimilation of satellite data: satellite data is distributed homogenously across the globe, and its use will increase, with the consequence that the imbalance in the amount of computations will decrease.

Figure 5 shows the elapsed time for one DA cycle for all experiments listed in Table 1. We can confirm that any resolution experiment could yield satisfactory scalability. This suggests that the new framework provides effective procedures for high-resolution and large-ensemble experiments on massively parallel computers.

## 5 Summary and discussion

In this paper, we proposed a framework that maintains data locality and maximizes the throughput of file I/O between the simulation model and the ensemble DA system. Each process manages data in a local disk. Separated parallel I/O is effective not only for read/write operations, but also for access to the metadata server. To reduce communication time, we changed the global communication of grid data to smaller group communication. The movement of data is strongly related to energy consumption as well as computational cost. Our approach is based on the concept of reducing the size and distance of moving data in the entire system. We assessed the performance of our framework on the K computer. Since the K computer is constructed as a distributed FS with a 3D mesh torus, it is not clear whether the approach proposed in this paper is effective with other FS and inter-node network topologies. However, the underlying concept—that minimizing data movement leads to better computational performance—will hold for most other supercomputer systems. This suggests that the cooperative design and development of the model and the DA system are necessary for optimization.

To improve I/O throughput for tentative use of disk storage, large HPC systems are being equipped with high-speed disks or non-volatile memory, such as a solid-state drive (SSD) with nodes (e.g., "Catalyst" at the Lawrence Livermore National Laboratory, TSUBAME2.5 in Tokyo Tech.). We propose using these "buffer disks" in future HPC systems. Each process occupies the local disk of its own node, and access collision can hence be avoided. We can also use the main memory as buffer storage depending on problem size. If memory is sufficient, we can facilitate the exchange of data between the



simulation model and the LETKF without any disk I/O. If we have a limited number of ensemble members to execute at the same time, we should use storage. We can store more than one set of output file to the buffer storage.

The observation-space data files used in this study were not distributed because these files were relatively small (e.g., < 1 MB for observation data, < 26 MB for sim-to-obs data involving 256 members). However, the amount of observational data

continues to increase. For example, massive multi-channel satellites, such as the Atmospheric Infrared Sounder (AIRS, Aumann et al, 2003), provide massive amounts of data for large areas. The geostationary satellite Himawari 8 generates approximately 50 times more data than its predecessor. Several hundred megabytes of observation data and several gigabytes of converted data by StoO are used in each assimilation instance. A node of a massive parallel supercomputer does not have sufficient memory to store these amounts of observation data. The time required by the master node to read such volumes of

data will become a bottleneck. We thus need to consider a division between observation data and their parallel I/O. The number of observational data items required by each process varies according to the number of divided parts of the simulation data and the spatial distribution of observation. Each process of the StoO program applies a conversion within the assigned area. By contrast, each process of the LETKF requires that the data be converted through multiple processes in the StoO according to the spatial localization range. Data exchanged using a library, such as MapReduce, is effective for such

altering many-to-many relationships. In order to increase the speed of data assimilation systems in the future, preprocessing of the observation data, such as dividing, grouping, and quality check, will be incorporated into our framework.

## 6 Code availability

The information of the NICAM can be found in the website (http://nicam.jp/). The source code of NICAM can be obtained upon request (see http://nicam.jp/hiki/?Research+Collaborations). The source code of the LETKF is open source and

available at https://code.google.com/archive/p/miyoshi/. The previous version of NICAM-LETKF DA system is based on this LETKF code and the tag version "NICAM.13" of the NICAM code. The new version of the DA system proposed in this study is based on tag version "NICAM.15", which includes LETKF code.

## Acknowledgments

This work was partially funded by MEXT's program for the Development and Improvement of Next-generation Ultra High-

Speed Computer System, under its Subsidies for Operating the Specific Advanced Large Research Facilities. The experiments were performed using the K computer at the RIKEN Advanced Institute for Computational Science. This study was partly supported by JAXA/PMM.

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

**Figure legends**

Figure 1. Schematic flow of the DA system with the LETKF.

Figure 2. Schematic diagram of the proposed framework in NICAM–LETKF.

Figure 3. The concept of grid division.

Figure 4. The time taken by NICAM–LETKF on the old and the new frameworks.

Figure 5. The time taken by NICAM–LETKF on the new framework.

**Table Captions**

Table 1. Configurations of DA experiment used to measure time taken on the K computer.






**Table 1. Configurations of DA experiment used to measure time taken on the K computer.**

| EXP. Name | Horizontal mesh size [km] | Number of vertical layers | Number of horizontal grids (per PE) | Number of horizontal grids (total) | Number of PE (per member) | Number of ensemble members | Number of PE (total) | Number of horizontal grids (per PE, shuffled) |
|---|---|---|---|---|---|---|---|---|
| G7R0E3 | 56 | 40 | 16900 | 169000 | 10 | 64 | 640 | 324 |
| G7R1E3 | 56 | 40 | 4356 | 174240 | 40 | 64 | 2560 | 100 |
| G7R2E3 | 56 | 40 | 1156 | 184960 | 160 | 64 | 10240 | 36 |
| G7R0E4 | 56 | 40 | 16900 | 169000 | 10 | 256 | 2560 | 100 |
| G7R1E4 | 56 | 40 | 4356 | 174240 | 40 | 256 | 10240 | 36 |
| G6R0E3 | 112 | 40 | 4356 | 43560 | 10 | 64 | 640 | 100 |
| G6R1E3 | 112 | 40 | 1156 | 46240 | 40 | 64 | 2560 | 36 |
| G8R1E3 | 28 | 40 | 16900 | 676000 | 40 | 64 | 2560 | 324 |
| G8R2E3 | 28 | 40 | 4356 | 696960 | 160 | 64 | 10240 | 100 |





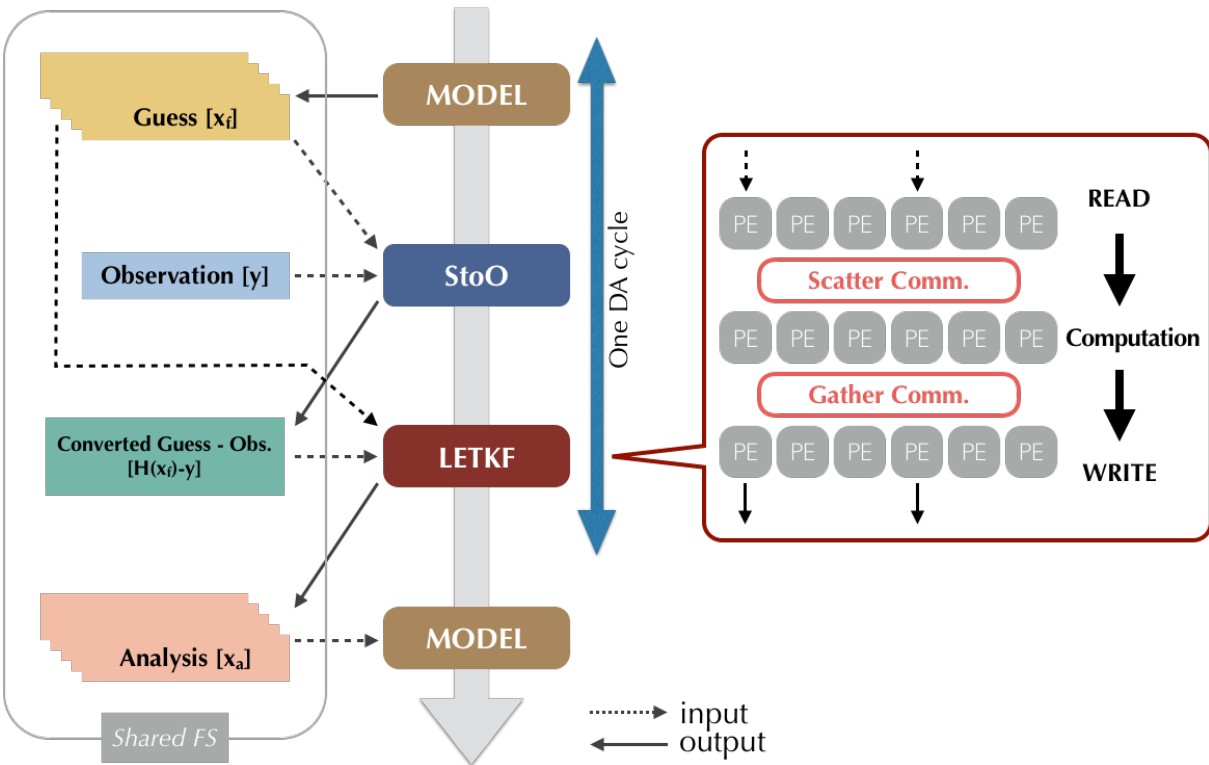

Figure 1. Schematic flow of the DA system with the LETKF.





Figure 2. Schematic diagram of the proposed framework in NICAM–LETKF.




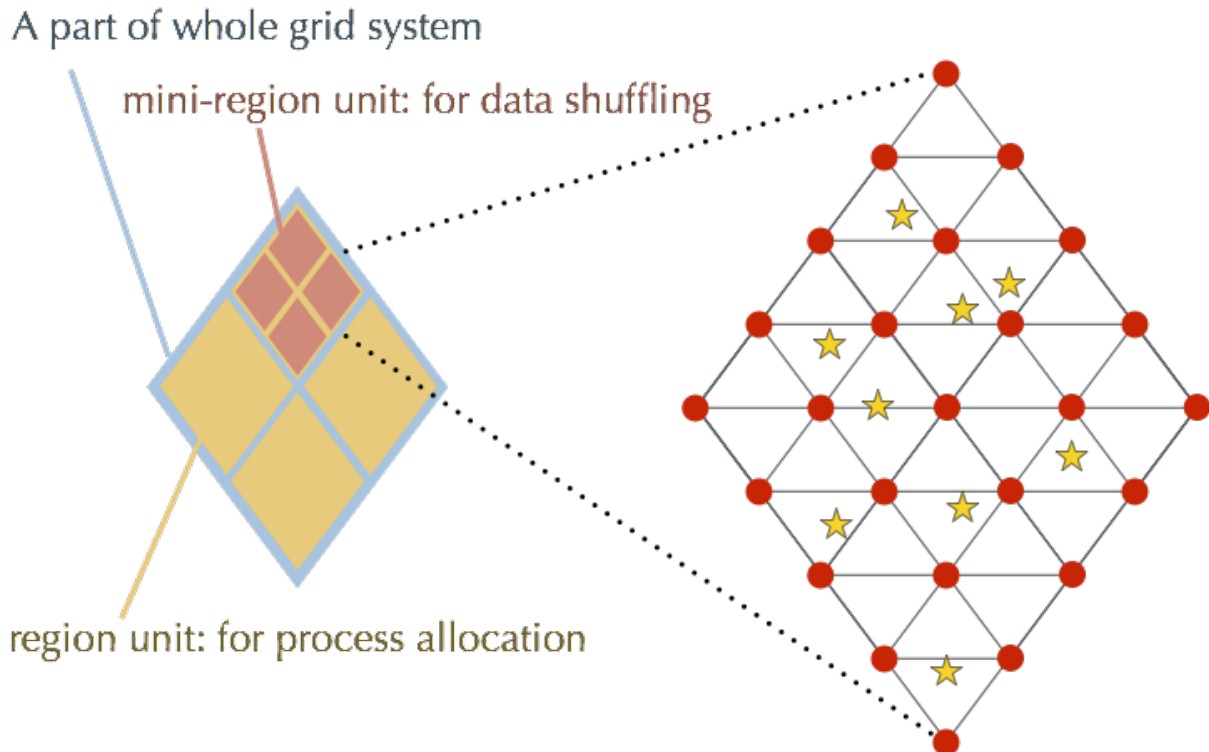

Figure 3. The concept of grid division.





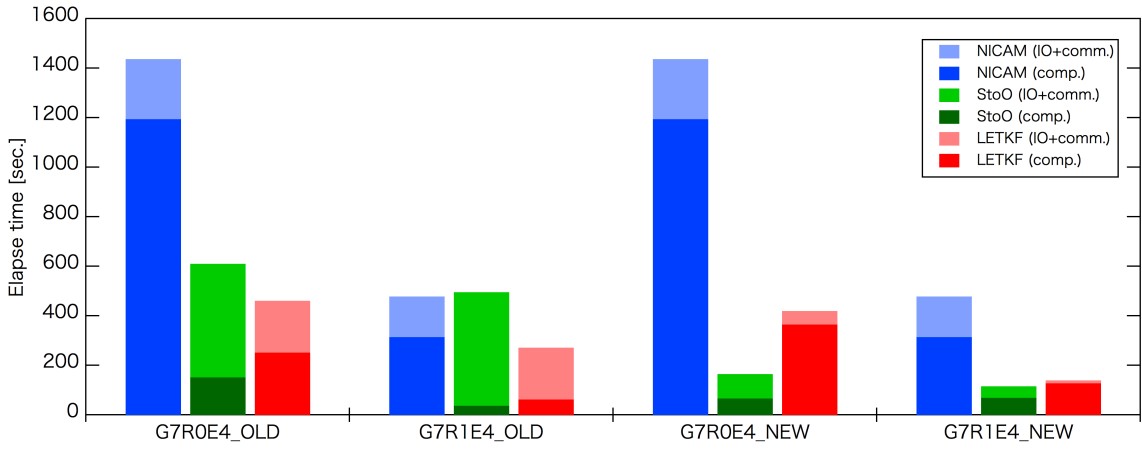

Figure 4. The time taken by NICAM–LETKF on the old and the new frameworks.

20



Figure 5. The time taken by NICAM–LETKF on the new framework.