# Peer review of "Performance evaluation of throughput-aware framework for ensemble data assimilation: The case of NICAM-LETKF"

_Geoscientific Model Development, 2016_

## Referee Comment (RC1) · Anonymous Referee #1 · 15 Apr 2016

The paper describes the design and technical implementation of an EnKF DA system for weather prediction at high spatial resolution and with a large ensemble size. The discussion specifically tackle the input/output and inetr-node communication in this setup. As these are aspects not often discussed in the literature and are important concerns for this type of applications, I believe the paper is a valuable contribution, especially for the operational data assimilation community. I think the paper could be further strengthened by addressing two aspects: 1) The paper does not seem to reference enough previous work in the scalability problems of EnKF systems at high resolution/ensemble size; 2) An important aspect for this type of applications are the load balancing issues, especially in connection with inhomogeneous observing systems.

---

## Referee Comment (RC2) · Anonymous Referee #2 · 25 Apr 2016

I agree with Anonymous Referee #1 that the paper treats an important subject and provides a valuable contribution to the community. The presentation is reasonably clear and comprehensive. Some specific comments:

1/ It seems from Figure 4 that the biggest gain going from the old to the new setup is for StoO. Could this gain not have been accomplished by integrating the StoO code into the model (perhaps with a coupler)? Both model and StoO are independent between different ensemble members.

2/ The timings presented are without any user output what so ever as far as I can see. In a operational scenario you would assume that at least the final analysis need to be output in a form that is suitable for producing maps,verification and to provide initial

states for an ensemble forecast. This would entail moving the data from local storage to global and "gluing" together the parts of the globe. The time this takes and its scaling behaviour would also need to taken into account.

3/ Related to 2/ above is the issue of resilience of the cycling system. If one of the nodes on which this assimilation system is cycling crashes and its data is lost we need somewhere to have a backup from which the data can be restored on a different node.

4/ One thing that is not clear in the presentation is the amount of observations used and if this is representative of today's operational data assimilations schemes (typically of the order of 10.000.000 observations/6 hour period). It is possible, if this number is much lower in the experiments presented, the issue of load-balancing , acknowledged by the authors, might become a much more important factor. The statement that the observation coverage will become homogeneous with the introduction of more satellite platform is dubious. This may be true in a time averaged sense, say over a day, but may not be true if the observation window is reduced to let say 3 hours in order to produce more frequent forecasts.
* * *

---

## Author Comment (AC1) · 30 May 2016

We appreciate your careful reviewing and positive comments and encouragements. Point-by-Point replies to your comments are as follows.

1) The paper does not seem to reference enough previous work in the scalability problems of EnKF systems at high resolution/ensemble size

[reply]

Thank you for the suggestion. We add some references.

2) An important aspect for this type of applications are the load balancing issues, es-

pecially in connection with inhomogeneous observing systems.

[reply]

There is a trade-off between computational load balancing and data movement. Miyoshi et al. (2010) changed grid point allocation to each node while paying attention to the load balancing. In our study, we revert this change to avoid the data movement by global communication. The ratio of floating point operations to memory accesses in LETKF analysis is large. This type of calculation is expected to become faster by performance enhancement of the future processor. It is easier than the improvement of throughput in global communication with massive nodes. We should select the best method for load barancing according to the number of nodes, the number of observation, the analytical method used in the LETKF, and the performance of computer system.

---

## Author Response (AR1)

We appreciate Anonymous Referee #1 and #2 for your careful reviewing and positive comments and encouragements. We revised the manuscript following your comments. Point-by-Point replies to your comments are as follows.

**5 Reply Comments for referee #1**

1) The paper does not seem to reference enough previous work in the scalability problems of EnKF systems at high resolution/ensemble size

[Reply]

Thank you for the suggestion. We add some references.

[Changes]

15 We add following sentences (p. 2, l. 30);

Humrud et al. (2015) have pointed out the limitation of scaling by using file I/O in the European Centre for Medium-range Weather Forecasts' (ECMWF) semi-operational Ensemble Kalman filter (EnKF) system. On the contrary, Houtekamer et al. (2014) showed satisfactory scalability in a Canadian operational EnKF DA system by using parallel I/O.

2) An important aspect for this type of applications are the load balancing issues, especially in connection with inhomogeneous observing systems.

25 [Reply]

There is a trade-off between computational load balancing and data movement. Miyoshi et al. (2010) changed grid point allocation to each node while paying attention to the load balancing. In our study, we revert this change to avoid the data movement by global communication. The ratio of floating point operations to memory accesses in LETKF analysis is large. This type of calculation is expected to become faster by performance enhancement of the future processor. It is

30 easier than the improvement of throughput in global communication with massive nodes. We should select the best method for load barancing according to the number of nodes, the number of observation, the analytical method used in the LETKF, and the performance of computer system.

[Changes]

We add following sentences (p 7, l. 19);

No satellite observation was used in this study. The number of observations will increase by one or two orders of magnitude if we use satellite observations. When high-resolution data assimilation is conducted with frequent assimilation cycles (e.g., three hours for the assimilation window), the inhomogeneity of the observation becomes larger and the load-balancing issue more critical. There is a trade-off between computational load balancing and data movement. It is worth considering the balancing technique described by Humrud et al. (2015). On the other hand, we argue that the speedup in computation in the LETKF is easier than that of global communication with massive nodes. The ratio of floating point operations to memory accesses in LETKF analysis is large. This type of calculation is expected to become faster by performance enhancement in future processors. We should select the best method for load balancing according to the number of nodes used, the number of observations, the analytical method used in the LETKF, and the performance of computer system.

**Reply Comments for referee #2**

1) It seems from Figure 4 that the biggest gain going from the old to the new setup is for StoO. Could this gain not have been accomplished by integrating the StoO code into the model (perhaps with a coupler)? Both model and StoO are independent between different ensemble members.

[Reply]

In this study, the program of StoO is separated from the atmospheric model and is executed individually. The gain of the StoO from old to new setup is mainly come from the improvement of poor I/O throughput and the collision of I/O requests. Following Miyoshi et al. (2010), multiple time slots of the observation and model output are used for StoO calculation in the data assimilation system of this study. The amount of model data inputted in the StoO is 7 times larger than that in the LETKF. For the LETKF, elapse times of both file I/O and MPI communication are decreased in the new framework.

[Changes]

We add following sentences (p 7, l. 5);

In particular, a significant reduction was observed in the time needed for StoO. In this study, multiple time slots of the observation and the model output were used for StoO calculation following Miyoshi et al. (2010). Thus, input data size

in the StoO was seven times larger than that in the LETKF. The improvement in I/O throughput largely contributed to the performance gain in StoO.

5   2) The timings presented are without any user output what so ever as far as I can see. In a operational scenario you would assume that at least the final analysis need to be output in a form that is suitable for producing maps,verification and to provide initial states for an ensemble forecast. This would entail moving the data from local storage to global and "gluing" together the parts of the globe. The time this takes and its scaling behaviour would also need to taken into account.

[Reply]
We agree that the scalability should be discussed according to the operational scenario. We did not take the time for post-processes of data assimilation cycle into account. This is because that such processes will be executed on the other nodes. We can choose appropriate time to copy the local data to global file system and/or the other node. Thus, the post-

15  processes do not block the sequential cycle of the data assimilation.
The post-processes should be executed by the multiple nodes. Remapping of the grid system is one of the most time-consuming part in the post-processes. We remap the variables from the icosahedral grid to the geodesic grid. This process takes similar time to the StoO program. Initial data for an ensemble forecast, which usually has higher spatial resolution than the ensemble analysis, is also prepared by remapping.

3) Related to 2/ above is the issue of resilience of the cycling system. If one of the nodes on which this assimilation system is cycling crashes and its data is lost we need somewhere to have a backup from which the data can be restored on a different node.

25

[Reply]
As discussed in the reply of comment 2), we can copy the data as a background process from the local storage to the global storage at the time when the data files are not busy. If the data assimilation cycle is stopped due to the node failure, we can restart the cycle by using the latest dataset in the global storage.

30  [Changes]
We add following sentences (p 8, l. 17);

The resilience of the DA cycle is also an important issue. It is better to backup analysis data during the cycle. We can copy the data as a background process from the local FS to the global FS when the data files are not busy. If the DA cycle is stopped due to the node failure, we can restart by using the latest data in the global FS.

4) One thing that is not clear in the presentation is the amount of observations used and if this is representative of today's operational data assimilations schemes (typically of the order of 10.000.000 observations/6 hour period). It is possible, if this number is much lower in the experiments presented, the issue of load-balancing , acknowledged by the authors, might become a much more important factor. The statement that the observation coverage will become

10 homogeneous with the introduction of more satellite platform is dubious. This may be true in a time averaged sense, say over a day, but may not be true if the observation window is reduced to let say 3 hours in order to produce more frequent forecasts.

[Reply]

15 Thank you for the valuable comment. We agree what you pointed out. When we conduct high-resolution data assimilation with frequent assimilation cycle, the inhomogeneity of the observation become larger. This is true even if we use satellite observations. We explain it in revision. The amount of the observation data per 6 hour period in this study was about 50,000. This is the number after the quality check and thinning from 1,000,000 observations.

We received similar comment from referee #1. There is a trade-off between computational load balancing and data

20 movement. The best setting is also depends on the machine. The number of observation and model resolution in this study will be insufficient to evaluate the future operational DA system. We will push forward the performance study of dynamic load balancing such as the technique described in Humrud et al. (2015) in the future. On the other hand, we argue that the speedup in computation in the LETKF is easier than that of global communication with massive nodes. The ratio of floating point operations to memory accesses in LETKF analysis is large. This type of calculation is

25 expected to become faster by performance enhancement in future processors.

[Changes]
We add the specification of the number of observation used (p 6, l. 22);

(50,000 per six hours on average)

We add following sentences (p 7, l. 19);

[revised manuscript text omitted]

**CERTIFICATE OF**
**ENGLISH EDITING**

This document certifies that the paper listed below has been edited to ensure that the language is clear and free of errors. The logical presentation of ideas and the structure of the paper were also checked during the editing process. The edit was performed by professional editors at Editage, a division of Cactus Communications. The intent of the author's message was not altered in any way during the editing process. The quality of the edit has been guaranteed, with the assumption that our suggested changes have been accepted and have not been further altered without the knowledge of our editors.

**TITLE OF THE PAPER**

Performance evaluation of throughput-aware framework for ensemble data assimilation: The case of NICAM-LETKF

**AUTHORS**

H. Yashiro, K. Terasaki, T. Miyoshi and H. Tomita

**JOB CODE**

KVZVM_1

[Figure]

Signature

Nikesh Gosalia,
Vice President, Author Services, Editage

Date of Issue
**May 30, 2016**

Editage, a brand of Cactus Communications, offers professional English language editing and publication support services to authors engaged in over 500 areas of research. Through its community of experienced editors, which includes doctors, engineers, published scientists, and researchers with peer review experience, Editage has successfully helped authors get published in internationally reputed journals. Authors who work with Editage are guaranteed excellent language quality and timely delivery.

[Figure]

**Contact Editage**

| Worldwide | Japan | Korea | China | Brazil | Taiwan |
|---|---|---|---|---|---|
| request@editage.com | submissions@editage.com | submit-korea@editage.com | fabiao@editage.cn | inquiry.brazil@editage.com | submitjobs@editage.com |
| +1 877-334-8243 | +81 03-6868-3348 | 1544-9241 | 400-005-6055 | 0800-892-20-97 | 02 2657 0306 |
| www.editage.com | www.editage.jp | www.editage.co.kr | www.editage.cn | www.editage.com.br | www.editage.com.tw |